# In Vitro and In Vivo Bioequivalence Study of 3D-Printed Instant-Dissolving Levetiracetam Tablets and Subsequent Personalized Dosing for Chinese Children Based on Physiological Pharmacokinetic Modeling

**DOI:** 10.3390/pharmaceutics14010020

**Published:** 2021-12-22

**Authors:** Xianfu Li, En Liang, Xiaoxuan Hong, Xiaolu Han, Conghui Li, Yuxi Wang, Zengming Wang, Aiping Zheng

**Affiliations:** 1State Key Laboratory of Toxicology and Medical Countermeasures, Beijing Institute of Pharmacology and Toxicology, 27th Taiping Road, Haidian District, Beijing 100850, China; xiaofu0924@163.com (X.L.); liangen0706@163.com (E.L.); hongxiaoxuan1216@163.com (X.H.); hanxiaolu921007@163.com (X.H.); angelina52819@163.com (C.L.); 2Department Pharmaceut, School Pharm, Yantai University, 32th Qingquan Road, Laishan District, Yantai 264005, China; 3Shanghai PharmoGo Co., Ltd., 3F, Block B, Weitai Building, No. 58, Lane 91, Eshan Road, Shanghai 200127, China; easonwang@pharmogo.com

**Keywords:** Binder jet 3D printing, physiological pharmacokinetic modeling, levetiracetam, bioequivalence, personalized dosing

## Abstract

Recently, the development of Binder Jet 3D printing technology has promoted the research and application of personalized formulations, which are especially useful for children’s medications. Additionally, physiological pharmacokinetic (PBPK) modeling can be used to guide drug development and drug dose selection. Multiple technologies can be used in combination to increase the safety and effectiveness of drug administration. In this study, we performed in vivo pharmacokinetic experiments in dogs with preprepared 3D-printed levetiracetam instant-dissolving tablets (LEV-IDTs). Bioequivalence analysis showed that the tablets were bioequivalent to commercially available preparations (Spritam^®^) for dogs. Additionally, we evaluated the bioequivalence of 3D-printed LEV-IDTs with Spritam^®^ by a population-based simulation based on the established PBPK model of levetiracetam for Chinese adults. Finally, we established a PBPK model of oral levetiracetam in Chinese children by combining the physiological parameters of children, and we simulated the PK (pharmacokinetics) curves of Chinese children aged 4 and 6 years that were administered the drug to provide precise guidance on adjusting the dose according to the effective dose range of the drug. Briefly, utilizing both Binder jet 3D printing technology and PBPK models is a promising route for personalized drug delivery with various age groups.

## 1. Introduction

Binder jet 3D printing (BJ-3DP) is a technology that constructs objects; in this process, digital models are first designed with the aid of a computer, then the objects are printed layer by layer, eventually turning the computer blueprint into a physical object [1]. Compared to traditional formulation manufacturing processes, 3D printing technology is extremely suitable for personalized drug delivery because it provides flexibility in adjusting drug doses, drug combinations, and production methods [2], especially for pediatric patients. Personalized doses can be printed and prepared according to individual requirements, which improves both the safety of children’s medication and pediatric patient compliance as pills can be printed with different shapes, colors, and flavors [3].

A physiologically based pharmacokinetic (PBPK) absorption model was developed in GastroPlus™ (Simulations Plus, Inc., Lancaster, PA, USA,) based on the physicochemical properties of the drug and human physiological characteristics [4,5,6,7]. Compared with the classical one-atrial chamber model and Wajima model, the PBPK model has greater predictive accuracy, which allows the drug concentrations in various tissues and organs throughout the body to be flexible by establishing a quantitative relationship between each organ and the plasma drug concentrations [8,9]. Jones et al. predicted human PK parameters and PK profiles for 19 compounds administered intravenously or orally using a PBPK model study based on preclinical data, and based on the measured values, approximately 70% of the compounds were accurately predicted [10].

In recent years, the PBPK model has rapidly developed from a subject in the academic community to a model in the field of drug discovery and development and has attracted extensive attention from drug regulatory authorities. In 2018, the European Medicines Agency (EMA) and the U.S. The Food and Drug Administration (FDA) formally issued specific guidelines for the PBPK platform and a relevant study report, which provided a basis for standardizing the application of the PBPK model to support new drug development [11,12].

The PBPK model is now widely used for guiding formulation development, human predictions [13], PK parameters, PK profiles [14,15,16], evaluations of bioequivalence, and clinical trial protocol design [17,18]. Pepin et al. developed a PBPK model for lesinurad (Zurampic^®^) in vivo by combining clinical trial data for intravenous and oral administration of lesinurad, which was used to evaluate the effect of in vitro dissolution of ZURAMPIC (lesinurad) tablets on in vivo performance, and they found that the key quality affecting the bioequivalence of clinical batches for this product involved particle size fractionation. With the PBPK model, Mitra et al. successfully predicted the bioequivalence of two batches of etoricoxib tablets manufactured at different sites, which could effectively reduce the bioequivalence (BE) cost for companies changing manufacturing sites [19].

In July 2015, the FDA approved the launch of Spritam^®^ by Aprecia, which was the first 3D-printed drug. Spritam^®^ tablets are printed layer-by-layer and molded on ZipDose’s technology platform with BJ-3DP technology. The instant-dissolving tablets (IDTs) prepared by this technology have a highly porous internal structure, which allows them to be rapidly dispersed in water, thus solving the swallowing difficulties patients experience. In this study, the BJ-3DP technology platform, which has the ability to produce Levetiracetam (LEV) dispersible tablets by batch printing, was built independently in the early stage, and the formulations produced by using the BJ-3DP technology platform have many advantages, such as large drug loading capacity, flexible dose, and rapid release [20,21].

Levetiracetam (LEV), the first marketed model drug for 3D-printed drug formulation, has ideal pharmacokinetic properties in humans. LEVis a second-generation antiepileptic drug that has many advantages including a broad antiepileptic spectrum, low adverse effects, good tolerability, and high safety, and is mainly used clinically for the adjunctive treatment of partial seizures in adults and children over 4 years old. LEV is characterized by an ideal pharmacokinetic profile including rapid oral absorption, a peak plasma concentration of approximately 1 h, oral bioavailability close to 100%, and a plasma protein binding rate <10% [22,23]. LEV has a low metabolic rate. In total, 93% of the administered dose is excreted after 24 h, 66% of the administered dose is unchanged in the urine, and 27% is excreted as an inactive metabolite [20].

In this study, we predicted the drug–time profile of 3D-printed LEV-IDTs in children by combining BJ-3DP and PBPK modeling, hoping to guide personalized drug delivery in children. We first performed in vivo pharmacokinetic experiments on 3D-printed LEV instant-dissolving tablets in beagles to demonstrate the bioequivalence of the formulation in beagles; moreover, we established a PBPK model for Chinese adults on GastroPlus™ software based on the physicochemical properties of LEV, and population-based simulations were performed to verify the bioequivalence of LEV-IDTs and Spritam^®^. Finally, a PBPK model for Chinese children was established based on the adult model, and the initial dose of LEV in Chinese children was predicted. This is the first time that the PBPK model was used to estimate the dose of LEV in children instead of their body weight, and the study offers a guideline for the individualized treatment of Chinese pediatric patients with epilepsy.

## 2. Materials and Methods

### 2.1. Materials

LEV-IDTs (250, 500, 750, and 1000 mg, Lab Printing); Spritam^®^ (50, 500, 750, and 1000 mg, Aprecia Pharmaceuticals Company, Blue Ash, OH, USA); dissolution meter (RC806D, Tianda Tianfa Technology Co., Ltd., Tianjin, China); LC-MS (LCMS-8060, SIMADZU, Kyoto, Japan); high-speed freezing centrifuge (ST16r, Thermo Fisher Scientific, Waltham, MA, USA).

### 2.2. Animals

The beagle had the following parameters: 15 dogs, weight: 10 ± 2 kg, sex: male, experimental animal certificate number: SCXK (Beijing) 2016-0001, purchased from Beijing Mars Biotechnology Co. Animal experiments followed the management regulations of the Experimental Animal Ethics Committee of the Military Medical Research Institute. (Project identification code: IACUC-DWZX-2020-691; date of approval: June 2020; name of the ethics committee: Animal Ethics Committee of the Military Medical Research Institute.)

### 2.3. In Vitro Dissolution Experiments

The USP II method and paddle method were used with 50 rpm; sampling times of 2.5, 5, 10, 15, 20, and 30 min; and a pH 1.2 hydrochloric acid solution as the media. The chromatographic conditions were as follows: octylsilane-bonded silica gel as an excipient; acetonitrile: buffer (10:90) (buffer: 1.4 g/L anhydrous disodium hydrogen phosphate solution, adjusted to pH 3.5 with phosphoric acid) as the mobile phase; detection wavelength: 205 nm; flow rate: 1.5 mL per minute; and column temperature: 30 °C.

### 2.4. In Vivo Dissolution Experiments

#### 2.4.1. Experimental Protocol and Sample Collection

The 15 beagles were divided into 5 parallel groups of 3 dogs each. All beagles were fasted (at least 12 h) before the test, and the drug was administered in the morning of the test day at a dose of 1 tablet per dog, and the dogs had empty stomachs. Food was given 4 h after administration, and throughout the experiment, the dogs were given free access to water. Blood samples were collected immediately before; 15, 30, 45, and 60 min after; and 1.5, 2, 4, 6, 8, 12, 24, and 48 h after administration of the dose of LEV.

Sample collection proceeded as follows: 0.5 mL of blood was collected from the forelimb vein of dogs using a 1 mL syringe in a sodium heparin tube, placed on ice, centrifuged (4000 rpm) for 10 min within 1 h after blood collection, and the plasma was separated and stored in a refrigerator at −40 °C to be frozen before subsequent testing.

#### 2.4.2. LC-MS/MS Conditions

The chromatographic conditions were as follows: the column was Kromasil 100-5 C18 (50 × 2.1 mm, AkzoNobel, Sundsvall, Sweden); the mobile phases were 0.1% formic acid-water (A) and 0.1% formic acid-acetonitrile (B) solution; the flow rate was 0.8 mL/min; the injection volume was 2 μL; and the column temperature was 40 °C. A gradient elution was used with the following time gradients: 0–0.5 min, (95% A, 5% B); 0.5–2.0 min, (95-5% A, 5–95% B); 2–2.5 min, (5% A, 95% B); 2.6–4min, (95% A, 5% B). The run time was 4 min.

An LC interface using pneumatic-assisted electrospray source ionization (ESI) conditions was utilized with a scanning mode: ion spray (IS) 4.0 KV, ion source temperature (TEM) 300 °C; drying gas flow: 10 L/min; heating gas flow: 10 L/min; nebulizing gas flow: 3 L/min; and collision energy (CE): −14 V.

#### 2.4.3. Sample Preparation

The blood concentration of LEV was determined by a validated high-performance liquid chromatography-mass spectrometry (HPLC-MS/MS, SIMADZU, Kyoto, Japan) method using propranolol as the internal standard and acetonitrile for direct protein precipitation.

The plasma sample of dogs (10 μL) was pipetted and added to 10 μL of blank acetonitrile, shaken and mixed, then 180 μL of acetonitrile containing internal standard (propranolol, 500 ng/mL) was added, vortexed and shaken, and centrifuged at 14,000 rpm for 10 min. Then, 10 μL of supernatant were taken and 90 μL of acetonitrile were added, mixed, and placed in the injection vial, and 2 μL of sample were injected for detection and analysis.

### 2.5. Pharmacokinetic Data Analysis

#### 2.5.1. Drug–Time Curve Plotting and Calculation of Pharmacokinetic Parameters

The concentrations of LEV in the plasma of dogs were measured separately at each sampling point, and the mean blood concentration–time profiles were plotted. The measured data were analyzed using Winnonlin 6.3 pharmacokinetic software to obtain the main pharmacokinetic parameters.

#### 2.5.2. Bioequivalence Evaluation in Dogs

ANOVAs were calculated separately for AUC_0–t_, AUC_0–∞_, and C_max_ for each specification formulation. The threshold for considering a significant difference was set at *p* ≤ 0.05, and two one-sided t-tests were used to assess whether the 90% confidence intervals (CIs) for the geometric mean ratios of C_max_, AUC_0–t_, and AUC_0–∞_ were within the range of 80.00–125.00%.

### 2.6. PBPK Modeling of LEV in Chinese Adults

All in vivo PK simulations in this study were performed in the PBPK model commercial software GastroPlus (Version 9.8, Simulations Plus, Inc., Lancaster, PA, USA). The physiologically based pharmacokinetic model was composed of a series of mathematical micro-equations. It describes the drug concentration in each organ using a single differential equation and then uses the overall differential equation to establish a quantitative relationship between the drug concentration and blood concentration in the organs, thus achieving a dynamic description of the drug concentration in organs and tissues throughout the body. The PBPK model was developed and validated as follows: First, we entered the physicochemical property parameters (e.g., molecular weight, solubility, pKa, LogP, LogD) and biopharmaceutical parameters (e.g., plasma free drug parameters) of LEV in Gatroplus software to obtain the initial PBPK model of LEV (Table 1). Then, the advanced compartmental absorption and transport (ACAT) model was used to describe the dissolution, precipitation, and absorption behavior of LEV in various sites of the gastrointestinal tract after drug administration. Next, to predict the distribution of LEV in the adult population, we used the Berezhkovskiy method in Gatroplus software to calculate the tissue-plasma partition coefficient (Kp) and volume of distribution (V_ss_) at steady state. The clearance rate for the adult model was obtained from the literature with a value of 0.96 mL/min/kg. Finally, we validated our model using data from clinical trials with different sizes and routes of administration of LEV reported in the literature (Table 2). Data from intravenous administration were used to validate clearance and distribution, and data from oral administration at different sizes from 250-1500 mg were used to validate absorption and dissolution. The model is accurate when the errors between the predicted and measured PK parameters (C_max_, AUC) for different dosing regimens are within 20% of the measured and predicted values. The formula for the absolute prediction error is shown in for Equation (1):(1)$$PE\%=\frac{\left|P\mathrm{redicted}-O\mathrm{bserved}\right|}{O\mathrm{bserved}}\times 100\%\;\left(PE:\;\mathrm{Absolute}\;\mathrm{prediction}\;\mathrm{error}\right)$$

### 2.7. Prediction of Bioequivalence

A population sample of Chinese individuals aged 25–50 was established using GastroPlus software (Version 9.8, Simulations Plus, Inc., Lancaster, PA, USA.) (50% male to 50% female, weight range: 54.33–67.91 kg, BMI range: 21.3–23.88). Combined with the dissolution curves of the reference formulation and the homemade formulation at pH 1.2, a crossover experimental design was utilized, and the same subjects were selected for population simulation and virtual bioequivalence experiments. The confidence interval of AUC and C_max_ between the test formulation and the reference formulation should be between 85% and 120%.

### 2.8. PBPK Modeling of LEV in Chinese Children

The PBPK model for children was built on the basis of the adult model, in which the input physicochemical parameters were kept consistent with the adult model. Physiological anatomical parameters were automatically calculated for Chinese children based on their height and weight in the age-related (PEAR) Physiology™ module of the Gatroplus software. The distribution parameters were calculated in the same way as in the adult model. Clearance rates were obtained from the literature with a value of 0.88 mL/min/kg. We performed in vivo PK prediction for Chinese children by using the model and compared the prediction results with the measured data. The measured data on the pharmacokinetics of oral LEV in Chinese children were obtained through a literature search [32].

### 2.9. The Application of the PBPK Model for Children in Personalized Medication and Dose Adjustment

Based on the established PBPK model for Chinese children, a PBPK model for Chinese children with different ages of oral LEV was established, and the relevant parameters were adjusted to establish PBPK models for Chinese children aged 4 and 6 years old to guide the individualized dosing of children. Based on the PBPK model for 6-year-old children, the model was simulated after a missed dose on the third night (60 h) of the seven-day dosing cycle, and the optimal remedial dose was predicted based on the model to guide the rationalization of medication administration after the missed dose. Some studies have pointed out that the reference range of clinically effective LEV blood concentrations for monotherapy of childhood epilepsy is 5 to 19 μg/mL, which is used to select the appropriate remedial dose to guide the rationalization of medication after missed doses [33].

## 3. Results

### 3.1. In Vitro Dissolution Experiment Results

The dissolution curve results (Figure 1) show that different sizes of LEV dispersible tablets have rapid release characteristics in pH 1.2 media, with more than 85% release in 2.5 min and complete release in 5 min. All of the above test results are similar for the various sizes of Spritam^®^.

### 3.2. Pharmacokinetic Study Results in Dogs

The in vivo pharmacokinetic results in dogs (Table 3 and Figure 2) showed that all sizes of 3D-printed LEV-IDTs had a rapid onset of action and reached C_max_ rapidly. The C_max_ of the homemade formulation 1000 mg and the original developed formulation 1000 mg were 118.89 and 127.85 μg/mL, respectively, with AUC_(0–∞)_ of 722.43 h × μg/mL and 721.98 h × μg/mL, indicating that there was no significant difference between the homemade and originally developed formulations of the same size.

### 3.3. Pharmacokinetic Study Results in Dogs

The in vivo pharmacokinetic results in beagle dogs (Table 3 and Figure 2) showed that the C_max_ and AUC_(0–∞)_ of different sizes of 3D-printed LEV-IDTs showed linear pharmacokinetic characteristics with increasing dose, and the half-life of LEV in vivo was about 3.2 h. All sizes of 3D-printed LEV-IDTs showed a rapid onset of action and reached T_max_ quickly. The C_max_ of 1000 mg was 118.89 and 127.85 μg/mL, respectively, and the AUC_(0–∞)_ was 722.43 h × μg/mL and 721.98 h × μg/mL, respectively, indicating that there was no significant difference between the homemade formulation and the original formulation of the same size.

### 3.4. Pharmacokinetic Study Results in Dogs

LEV is a BCS Class I drug, that is not restricted by age, weight, etc., and can be released rapidly in vivo; therefore, a parallel design test was chosen. ANOVA and two-way one-sided t-tests were performed for AUC_(0–∞)_, AUC_(0–t)_, and C_max_, respectively, and the results of 90% confidence interval analysis are shown in Table 4, with 90% confidence intervals of 93.42–107.85% for AUC_(0–∞)_, 93.23–109.09% for AUC_(0–t)_, and 90% confidence interval for C_max_ 99.35–116.81%, which are all in the range of 80–125%, demonstrating that LEV-IDTs were bioequivalent to Spritam^®^ in dogs.

### 3.5. Results of the Development and Validation of the PBPK Model for Chinese Adults

In the Chinese adult oral PBPK model, plasma partition coefficient (Kp) values for each tissue were calculated using the Berezhkovskiy method with a steady-state volume of distribution (V_ss_) of approximately 0.54 L/kg. The clearance was calculated using PK data from Chinese adults administered intravenously at approximately 0.063 L/h/kg, and a 66% clearance was defined as renal tissue. The predicted results of intravenous and oral PK in Chinese adults are shown in Figure 3.

The results showed that the predicted PK parameters (C_max_, AUC) for different dosing regimens were within a 20% error compared to the measured values using the LEV Chinese adult intravenous and oral PBPK models developed from the collected physicochemical and biopharmaceutical parameters, Chinese adult gastrointestinal model, physiological parameters, and disposition parameters. Part of the T_max_ results were in PE > 20%, which may be due to the problem of the blood collection point setting during the experiment. Overall, the currently established Chinese adult LEV PBPK model can accurately reflect the process of dissolution, absorption, distribution, and clearance of this drug in vivo. As shown in Figure 4, LEV is completely released in Chinese adults in 30 min, and the absorption seems to determine the rate of LEV entering the body circulation. As shown in Figure 5, the main absorption site of LEV was in the upper part of the small intestine, in which 49.7% was absorbed in the duodenum and 36.2% and 7.8% in jejunum 1 and jejunum 2, respectively, which was almost completely absorbed (Fa = 96.7%).

### 3.6. Results of Bioequivalence Evaluation Based on the PBPK Model

According to the calculated results (Figure 6), the T_max_ results for both LEV-IDTs and Spritam^®^ were 1.5 h. The mean C_max_ values for both were 24.76 and 25.39 μg/mL, respectively, with a 90% CI of 93.0% to 102.0%; the mean AUC_0–t_ values were 217.4 and 222.5 μg × h/mL, respectively, with a 90% CI of 91.1% to 104.3%. The mean values of AUC_0–t_ were 217.4 and 222.5 μg × h/mL, respectively, and the 90% CIs of the mean ratios were 91.1% to 104.3%. The results of BE prediction using T_max_, C_max_, and AUC_0–t_ as parameters showed that LEV-IDTs and Spritam^®^ were bioequivalent in humans.

### 3.7. Simulation Results of the LEV PBPK Model for Chinese Children

In the Chinese children’s oral PBPK model, Kp values for each tissue were also calculated using the Berezhkovskiy method [34] with a V_ss_ of approximately 0.61 L/kg. Clearance was calculated using clinical empirical PK data from Chinese children, which was approximately 0.053 L/h/kg, and a 66% clearance was defined as renal tissue. The predicted results of oral PK in the Chinese pediatric population (age: 0.5–15 years, weight: 4–87 kg, 55% male, dose: 10–60 mg/kg) are shown in Figure 7. The PK data of oral LEV in Chinese children predicted by this model almost entirely covered the actual clinical data after oral administration of LEV in Chinese children. Based on this model, the PK curves of oral LEV in Chinese children of different ages, weights, and dosing regimens can be predicted, which can be used for the formulation of dosing regimens and optimization of dosing in children. The PK data of oral LEV in Chinese children was predicted by using the ACAT model and tissue physiological parameters built into GastroPlus, combined with the information of actual clinical subjects and dosing regimens. Based on this model, the PK curves of oral LEV in Chinese children of different ages, weights, and dosing regimens can be predicted, which can be used for the formulation of dosing regimens and optimization of dosing in children.

### 3.8. Design of Individualized Doses for 4-Year-Old Chinese Children

According to the marketing instructions of Rohu LEV, the starting dose of LEV is 10 mg/kg twice daily (q 12 h) in 4-year-old children (weight ≤ 50 kg BMI ≤ 20); therefore, the drug–time profile of oral LEV 177.8 mg (q 12 h) in Chinese children was first simulated on the basis of the PBPK model in Chinese children. The blood concentration of LEV was found to fluctuate in the range of 3.08–12.75 μg/mL (Figure 8A), but the trough concentration was lower than the reference range of the clinically effective LEV blood concentration for monotherapy in children with epilepsy, which was 5~19 μg/mL. Adjusting the dose of administration to an initial dose of 300 mg, followed by 200 mg each time for simulation, it was found that the blood concentration of LEV could be maintained in the range of 5.02~15.98 μg/mL (Figure 8B), which was basically consistent with the clinically effective LEV blood concentration range. Therefore, to maintain a smooth and effective therapeutic concentration, the ideal dosing regimen for Chinese 4-year-old children with epilepsy could be 300 mg for the first dose, followed by 200 mg for each subsequent dose (q 12 h).

### 3.9. Design of Individualized Doses for 6-Year-Old Chinese Children

The pharmacokinetic profile of a 6-year-old Chinese child (weight 22.26 kg, BMI 16.07) after an oral administration of LEV was simulated at a dose of 222.6 mg (q 12 h) for seven consecutive days based on the dose required by the LEV instructions (10 mg/kg) to predict the PK profile of LEV in children. The valley concentration of the model results was only 4.13 μg/mL, which is below the recommended blood concentration range, so the initial dose was increased to 280 mg, and the subsequent dose was 222.6 mg per dose, so the blood concentration in pediatric patients was maintained more consistently at 5.22–14.80 μg/mL.

Children with epilepsy may have missed or late doses during long-term medication treatment, and missed doses need to be remedied in a timely manner. To ensure the stability of the blood concentration in the case of missed doses, this study explored the appropriate remedial dose based on the PBPK model in children. After two days of steady-state blood concentrations, the child missed a dose on the third night (time: 60 h) and then planned to take 300 mg of LEV in the morning of the fourth day (time: 72 h) for remediation, resuming 222.6 mg (q 12 h) after 84 h. With one missed dose at 60 h, the blood concentration was only 2.134 μg/mL at 72 h. Remediation with 300 mg LEV allowed the blood concentration to rapidly return to that of a normally administered dose and had no effect on subsequent dosing (Figure 9B). In the case of remedial doses of 275 or 350 mg (Figure 9C,D), although the blood concentration of 275 mg remedial increased at 72 h, it did not reach the blood concentration at normal dosing, and the steady-state blood concentration at normal dosing was exceeded at 350 mg remedial. To ensure the safety of drug administration, the appropriate remedial dose should be chosen to avoid adverse reactions caused by excessive remedial doses and poor therapeutic effects that occur when remediation is too low. A total of 300 mg LEV should be taken for remediation. The PBPK model for dose remediation in patients with missed doses is based on the prediction of blood concentration in the patient’s body, which has good accuracy and reliability, can effectively ensure the efficacy of patients taking long-term medication, and can avoid the adverse effects of taking a dose that is too high.

## 4. Discussion and Conclusions

According to criteria established by the FDA and generally applied by other regulatory agencies, a test preparation and a reference preparation are considered bioequivalent when the 90% confidence interval for the geometric mean ratios of C_max_, AUC_0–t_ and AUC_0–∞_ between the reference preparation and the test preparation are within the range of 80.00–125.00%. In this study, the bioequivalence between 3D-printed LEV-IDTs and Spritam^®^ was investigated using both in vivo pharmacokinetic experiments in beagles and virtual BE simulations of the human PBPK model. The results of both experiments showed that the 90% confidence intervals of the geometric mean ratios of C_max_, AUC_0–t_, and AUC_0–∞_ between LEV-IDTs and Spritam^®^ in both beagles and humans were in the range of 80.00–125.00%, indicating that laboratory-made LEV-IDTs could be bioequivalent to Spritam^®^ in living organisms. Additionally, a PBPK model for Chinese children aged 0.5–15 years was developed, and the model was used to provide feasible suggestions for individualized dosing regimens for children of different ages so that effective therapeutic concentrations could be achieved in different children.

The PBPK model can be used to build an accurate model to predict PK profiles and PK parameters of human drugs by combining the physicochemical properties of drugs and human physiological parameters. Stefan et al. used the PBPK model to simulate the pk profiles of 26 drugs, such as Lorcainide and Domperidone, after intravenous and oral administration, respectively [16]. The results showed that pharmacokinetic parameters, such as AUC, V_ss_, and C_max_, could be accurately predicted within 2-fold, 74%, 70%, and 65%, respectively [16]. In this study, physiological parameters were estimated for adult and pediatric subjects based on the built-in age-related (PEAR) Physiology™ module of GastroPlus, and PBPK models for different age groups were developed by combining information on the age and weight of actual clinical subjects. The predictive power of the model was further improved by using different routes of administration and different dose levels of LEV (Figure 3).

Zhang Fan et al. selected isosorbide mononitrate as a model drug and used the PBPK model to assess whether a rapid dissolution profile (≥85% in 30 min) is critical to ensure the bioequivalence of isosorbide mononitrate and to establish a clinically relevant dissolution specification for screening BE or non-BE drugs [17]. Our choice of LEV, which like isosorbide mononitrate is also a BCS class I drug, has the advantage of high solubility and high permeability, and its rate and extent of absorption are unlikely to depend on the drug dissolution and gastrointestinal transit time, with the theoretical rate-limiting step being gastric emptying. We compared the measured PK parameters (C_max_, AUC) of different clinical trials in the literature with the predicted values of our PBPK model and the errors were within 20% (PE < 20%), the T_max_ part of the results PE > 20%, which may be due to the problem of blood collection point setting during the experiment (Table 5).

Conventionally, pediatric drug doses have been adjusted based on adult doses using weight-, height-, or age-related functions. This simple anisotropic growth approach can be problematic when complex absorption and disposition processes are encountered and may not accurately predict exposure, especially in young children. Scaling by body surface area has been reported to lead to overdose in neonates and infants because of formula inaccuracy or the unpredictability of low drug-metabolizing enzyme activity at birth [35]. The PBPK model allows mathematical modeling of pediatric growth and organ maturation to predict the pharmacokinetic characteristics of drugs in children.

Both BJ-3DP technology and the PBPK model have demonstrated unique advantages in the field of drug development and manufacturing, and its digital and personalized manufacturing approach has injected new momentum and new models for the evolution of pediatric formulations in the current healthcare environment. The innovation of this article is the combination of BJ-3DP technology and the PBPK model, which provides a solution that can meet the personalized drug delivery needs of children of different ages. BJ-3DP technology can adjust the dose flexibly through the adjustment of the model size to achieve the preparation of different sizes of products; additionally, PBPK can predict the personalized drug delivery needs of different children according to their physiological parameters. In the future, if BJ-3DP technology can be combined with the PBPK model, after the patient is diagnosed, the pharmaceutical worker will input the physiological parameters of the patient through the PBPK model, and the model will predict the dose required for the ideal blood concentration and then transmit it to the BJ-3DP platform, which will print out the corresponding specification by adjusting the model, thus truly meeting the personalized drug delivery needs of different people.

## Figures and Tables

**Figure 1 pharmaceutics-14-00020-f001:**
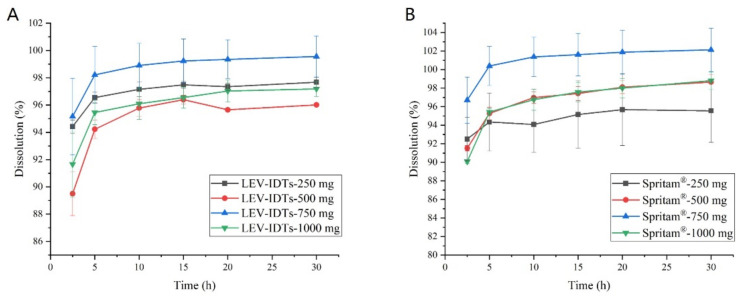
Dissolution curves of different sizes of 3D-printed (**A**) LEV-IDTs and (**B**) Spritam^®^ in pH 1.2 media (*n* = 6).

**Figure 2 pharmaceutics-14-00020-f002:**
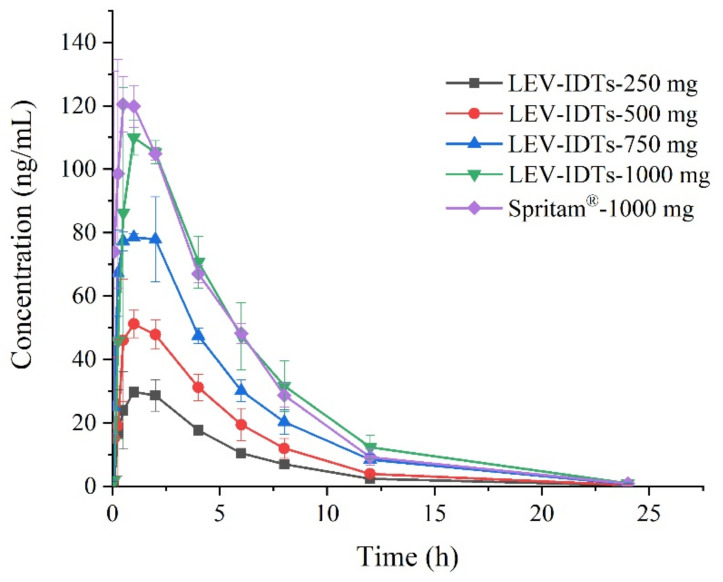
Drug–time profiles after oral administration of LEV in dogs (*n* = 3).

**Figure 3 pharmaceutics-14-00020-f003:**
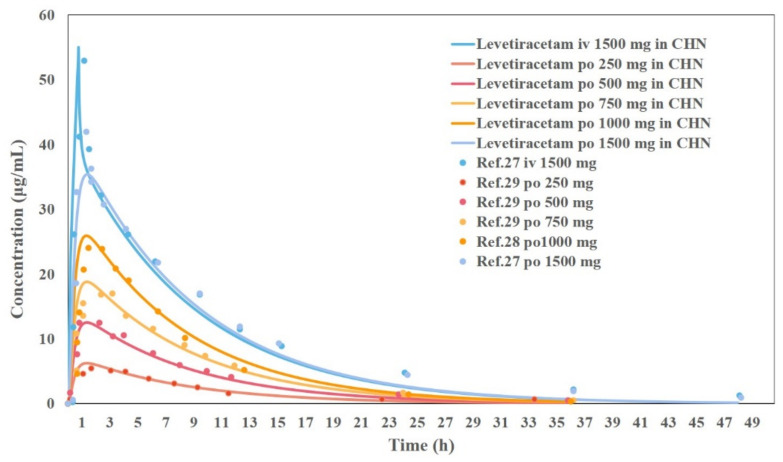
Validation results of the PBPK model with different dosing regimens (the solid line is the prediction result and the dashed line is the validation result).

**Figure 4 pharmaceutics-14-00020-f004:**
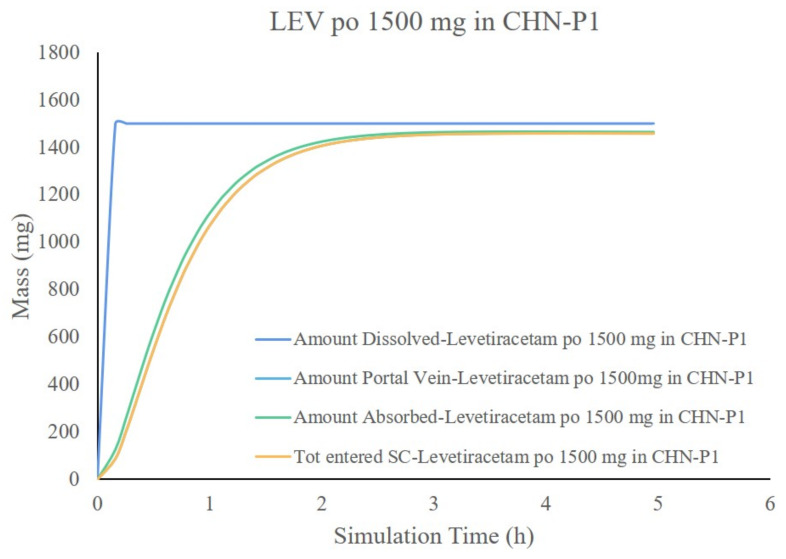
Dissolution and absorption curve of 1500 mg LEV orally in Chinese adults.

**Figure 5 pharmaceutics-14-00020-f005:**
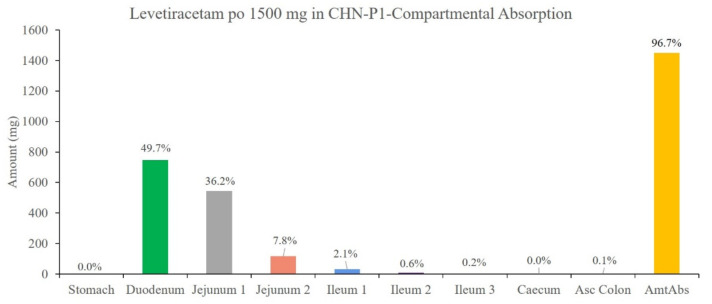
Percentage absorption of each intestinal segment of 1500 mg LEV orally in Chinese adults.

**Figure 6 pharmaceutics-14-00020-f006:**
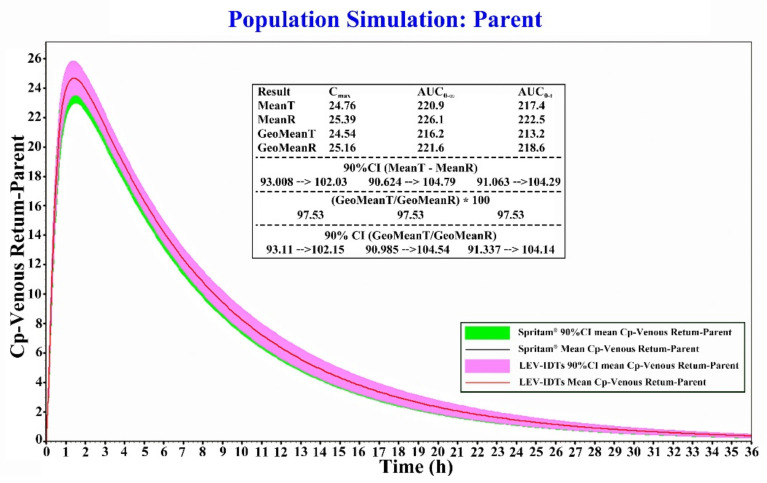
Results of LEV-IDTs and Spritam^®^ bioequivalence simulations.

**Figure 7 pharmaceutics-14-00020-f007:**
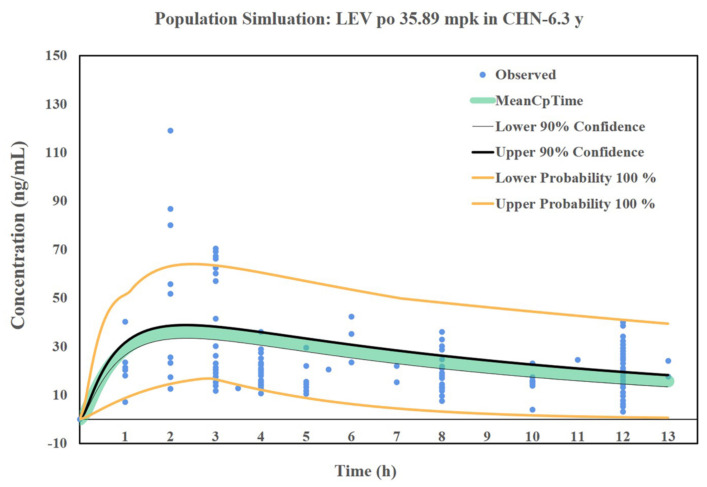
Prediction results of the oral LEV PBPK model in the Chinese pediatric population.

**Figure 8 pharmaceutics-14-00020-f008:**
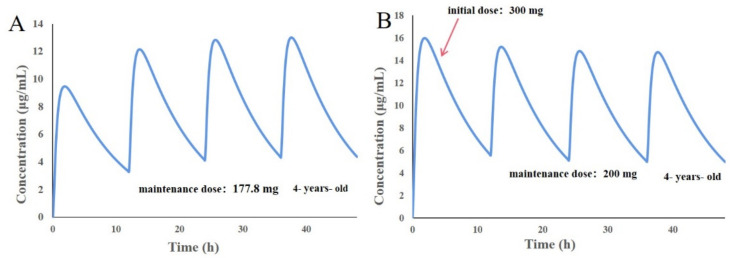
Predicted results of the oral PBPK model in 4-year-old Chinese children. (**A**) 177.8 mg, q 12 h; (**B**) 300 mg + 200 mg, q 12 h.

**Figure 9 pharmaceutics-14-00020-f009:**
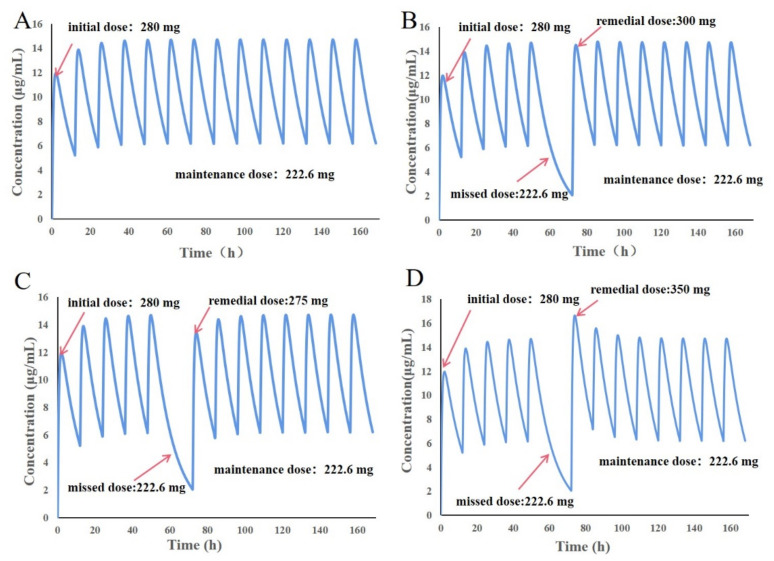
Prediction results of the 280 mg + 222.6 mg oral LEV-PBPK model in 6-year-old Chinese children. (**A**) Seven-day drug–time curve after normal administration; (**B**) 300 mg drug–time curve after remediation of a missed dose; (**C**) 275 mg drug–time curve after a missed dose; (**D**) 350 mg drug–time curve after remediation of a missed dose.

**Table 1 pharmaceutics-14-00020-t001:** Physicochemical and biopharmaceutical parameters of LEV.

Parameters	Values	References
Relative molecular mass	170.21 g/mol	Gatroplus
LogP	0.8	[24]
pKa	16.1	[24]
LogD	0.9	[24]
Rbp	1.11	Gatroplus predict
Water solubility	1.04 g/mol	[25]
Fup	97%	[26]
Intestinal permeability coefficient	9.6 × 10^−6^ cm/s	[27]
Adult clearance rate	0.96 mL/min/kg	[22]
Clearance rate of children in China	0.88 mL/min/kg	[28]
V_d_	0.5~0.7 L/kg	[22]

**Table 2 pharmaceutics-14-00020-t002:** In vivo pharmacokinetic literature information summary for LEV.

Administration	Dose (mg)	Quantities	Age	Weight (kg)	BMI	References
IV	1500 mg	24	23.71	64.19 ± 7.38	--	[29]
Oral tablet	1500 mg	24	23.71	64.19 ± 7.38	--	[29]
Oral tablet	1000 mg	8	33.8	57.4 ± 3.70	22.1 ± 0.8	[30]
Oral tablet	750 mg	8	21.0	58.8 ± 6.50	19~24	[31]
Oral tablet	500 mg	8	21.0	58.8 ± 6.50	19~24	[31]
Oral tablet	250 mg	8	21.0	58.8 ± 6.50	19~24	[31]

**Table 3 pharmaceutics-14-00020-t003:** Key pharmacokinetic parameters after oral administration of LEV in dogs (*n* = 3).

Parameters	Formulation				
	LEV-IDTs-250 mg	LEV-IDTs-500 mg	LEV-IDTs-750 mg	LEV-IDTs-1000 mg	Spritam^®^
HL_Lambda_z (h)	3.54 ± 0.32	3.36 ± 0.27	3.30 ± 0.32	3.14 ± 0.31	3.13 ± 0.46
T_max_ (h)	1.00 ± 0.87	1.00 ± 0.87	1.67 ± 0.58	0.83 ± 0.29	0.28 ± 0.21
C_max_ (μg/mL)	32.01 ± 1.88	55.54 ± 4.83	82.81 ± 5.14	118.89 ± 11.07	127.85 ± 7.09
AUC_(0–t)_ (h × μg/mL)	176.16 ± 1.09	304.95 ± 31.34	510.01 ± 10.39	718.05 ± 93.24	717.68 ± 48.66
AUC_(0–∞)_ (h × μg/mL)	177.82 ± 0.66	307.17 ± 32.22	513.44 ± 9.60	722.43 ± 94.59	721.98 ± 51.03
MRT_last_ (h)	4.68 ± 0.08	4.58 ± 0.48	4.81 ± 0.43	5.02 ± 0.40	4.52 ± 0.33

**Table 4 pharmaceutics-14-00020-t004:** Results of bioequivalence evaluation.

Parameters.	Lower Limit	Upper Limit	Standard
AUC_(0–∞)_	93.42%	107.85%	80–125%
AUC_(0–t)_	93.23%	108.09%	
C_max_	99.35%	116.81%	

**Table 5 pharmaceutics-14-00020-t005:** Statistical table of prediction results.

		IV 1500 mg	Oral 250 mg	Oral 500 mg	Oral 750 mg	Oral 1000 mg	Oral 1500 mg
C_max_	Observed	50.80	5.51	12.50	17.00	24.10	35.30
	Predicted	54.83	6.268	12.54	18.80	25.90	35.43
	PE	7.93	13.76	0.32	10.59	7.47	0.37
AUC_0–t_	Observed	367.20	62.56	138.50	189.80	222.80	402.30
	Predicted	357.40	58.96	117.90	176.90	236.60	354.70
	PE	2.67	5.75	14.87	6.80	6.19	11.83
AUC_0–inf_	Observed	370.70	72.56	145.40	196.10	226.40	408.30
	Predicted	358.90	59.75	119.50	179.30	239.40	356.30
	*PE*	3.18	17.65	17.81	8.57	5.74	12.74
T_max_	Observed	0.711	1.67	0.80	3.16	1.45	0.646
	Predicted	0.75	1.32	1.32	1.32	1.30	1.38
	*PE*	5.49	20.96	65.00	58.23	10.34	113.62

(C_max_: μg/mL; AUC_0–t_: μg × h/mL; AUC_0–inf_: μg × h/mL; T_max_: h).

## Data Availability

The data presented in this study are available on request from the corresponding author. The data are not publicly available due to some privacy issues about drug development.

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
