# Peer review of "In Vitro and In Vivo Bioequivalence Study of 3D-Printed Instant-Dissolving Levetiracetam Tablets and Subsequent Personalized Dosing for Chinese Children Based on Physiological Pharmacokinetic Modeling"

_pharmaceutics, 2021, doi:10.3390/pharmaceutics14010020_

Round 1

Reviewer 1 Report

The manuscript presents a new method for preparation of dosage forms which can be used for precise dosing of the drugs. The application of PBPK models in the prediction of the efficacy and the determination of dosing regimens was discussed. These approaches were evaluated in the levetriacetam administration in Beagles and in adult patients. Further, based on population approach, a PK model for children was established and initial dose was suggested. The steps in the study and subsequent application the PK tools are correct and can lead to achievement of the goal of the study.

M & M

More detailed description is required for PK analysis. Which method and which model were used for analysis of the data?

Page 5/15, Line 188: Please revise/explain “a population-based population”. The meaning is not clear.

Results

Minor issue, but please, use capital letter at the beginning of the heading

Fig. 2: It is strange to have negative values for concentration at time 0 h. Please, check again the original data and correct if you use concentration< LOQ.

Page 8/15, Line 261: “dissolution” was written twice.

Table 5: PE is not explained under the table. Please, add the information.

The formula for absolute prediction error should be placed in material and methods.

Discussion needs a revision. It will be of great value if the authors compare the success of their model to the results obtained for other drugs. The authors outlined the limitations of the approach bur it will be nice if they clearly state that this is not the case of levetiracetam. It would be nice if the authors are able to compare the predicted concentrations with the measured concentrations from the literature data of the drug. This will give more strength of the manuscript when the application of PBPK model is discussed.

Reviewer 2 Report

The article “In vitro and in vivo bioequivalence study of 3D printed instant dissolving levetiracetam tablets and subsequent personalized dosing for Chinese children based on physiological pharmacokinetic modeling” is on a very exciting application of physiology-based pharmacokinetic modeling. Following are my comments/suggestions,
1. Please include information related to the metabolism/clearance of levetiracetam in the introduction section.
2. Kindly explain how the PBPK model was developed? How many data sets were used for model building and how many data sets were used for subsequent model verification? To clear this point please see the following references,
i. Physiologically based pharmacokinetic (PBPK) modeling and simulation approaches a systematic review of published models, applications, and model verification. Drug Metabolism and Disposition. 2015 Nov 1;43(11):1823-37.
ii. A Physiologically Based Pharmacokinetic Model for Predicting Diazepam Pharmacokinetics after Intravenous, Oral, Intranasal, and Rectal Applications" Pharmaceutics 13, no. 9: 1480. https://doi.org/10.3390/pharmaceutics13091480
3. Kindly provide some details about the developed PBPK model structure, what was the absorption model? What was the distribution model?
4. In the prediction of bioequivalence, what do authors mean by population-based population?
5. In PBPK modeling of LEV in Chinese children, it is written that “ The PBPK model for Chinese children was developed based on the PBPK model for adults combined with clinical subject information (age, height, weight, etc.), dosing regimen, and pharmacokinetic data for oral LEV in Chinese children obtained from a literature search” how was this literature search conducted?
6. In the section “The application of the PBPK model for children in personalized medication and dose adjustment” please explain how the population for Chinese children was created? What population-specific parameters were included for assigning variability in the created virtual population?
7. Figure 1 is not clear, and it is very hard to follow the shown data, kindly make it clear and readable.
8. Figure 3 is not clear and it is difficult to follow the shown data.
9. To see the predicted results clearly, kindly change the y-axis of Figure 7 to log values, as it seems the predictions are not in accordance with the observed data.
10. Figures 8 and 9 are not clear and are very hard to follow.

Reviewer 3 Report

This manuscript introduce the synthesis of bioequivalence study of 3D printed instant-dissolving levetiracetam tablets and subsequent personalized dosing for Chinese children based on physiological pharmacokinetic modeling. The authors performed appropriate and in vivo test to evaluate instant-dissolving levetiracetam tablets. So, it is suitable for publication in the journal "Pharmaceutics" since it has the interesting topic and results. However, it has the following revised parts. They should be checked prior to the publication. Followings are recommended for the revision.

Major revisions

  1. All ‘in vitro’ and ‘in vivo’ should be written in italics. (ex. in vitro, in vivo)
  2. The quality of all figures should be increased, and the font size should be increased so that the characters in the figure can be seen clearly.
  3. All spritam ® must be written in superscript. (ex. spritam®)
  4. In page 2, line 74-100, since the connection of content between paragraphs is not smooth, it should be written more smoothly.
  5. In page 3, line 108, 120 the content and title do not match and need to be corrected.
  6. In page 3, line 138-139, it is not stated what percentage of mobile phase A enters in a few minutes. You should write in detail how many percentages A and B enter each time period.

Minor revisions

  1. In page 1, line 28, before using the PK abbreviation, write the full name of pharmacokinetics first, and then use the abbreviation.
  2. In page 2, line 64, Spaces between ‘]’ and ‘,’ must be removed.
  3. In page 3, line 114, there is a grammar error. (was -> were)
  4. In page 3, line 130, the ‘-’ in ‘cen-tri’ must be removed.
  5. In page 4, line 148-152, a space is required between numbers and units. (10μl ->10 μl)
  6. In page 5, line 189, the ‘y’ should be deleted.
  7. In page 6, line 226, the ‘-’ in ‘be-tween’ must be removed.
  8. In page 6, line 229, and in page 5, line 186, a space is required between numbers and units in table 2, 3. (ex. 3.54±0.32 -> 3.54 ± 0.32)
  9. In page 8, line 261, the letters of ‘dissolution’ are duplicated and should be deleted.
  10. In page 9, line 279, In ‘AUC0-t’, ‘0-t’ should be placed as a subscript. (AUC0-t -> AUC0-t)
  11. In page 10, line 292, In ‘Vss’, ‘ss’ should be placed as a subscript. . (Vss -> Vss)
  12. In page 10, line 301, there is a grammar error. (were -> was)
  13. In page 10, line 308, you should be a space between units and numbers in the figure 7 title. (35.89mpk -> 389 mpk)
  14. In page 11, line 315, as shown in line 331 below, the BMI index must be added next to kg.
  15. In page 11, line 315, in ‘q 12 h’, you should explain what q is or write the full name of the abbreviation.
  16. `Figure` and its abbreviated form, `Fig`, are mixed throughout the present text. Please unify them.
  17. In page 11, line 329, and in page 12, line 360, spaces are required between letters and numbers. (300 mg +200 mg -> 300 mg + 200 mg)
  18. In page 12, line 363, you should be deleted ‘)’.
  19. In page 13, line 416, ‘References’ should be numbered. And the reference format should be indented the same as the paragraph format above.

Round 2

Reviewer 2 Report

The authors have addressed all of my comments/suggestions in their revised submission.